# Directly imaging emergence of phase separation in peroxidized lipid membranes

Miguel Paez-Perez [1], Aurimas Vyšniauskas [1,2], Ismael López-Duarte [1,3], Eulalie J. Lafarge [4], Raquel López-Ríos De Castro [5], Carlos M. Marques [4,6], André P. Schroder[4,7], Pierre Muller [4], Christian D. Lorenz [5], Nicholas J. Brooks [1✉] & Marina K. Kuimova [1✉]

Lipid peroxidation is a process which is key in cell signaling and disease, it is exploited in cancer therapy in the form of photodynamic therapy. The appearance of hydrophilic moieties within the bilayer's hydrocarbon core will dramatically alter the structure and mechanical behavior of membranes. Here, we combine viscosity sensitive fluorophores, advanced microscopy, and X-ray diffraction and molecular simulations to directly and quantitatively measure the bilayer's structural and viscoelastic properties, and correlate these with atomistic molecular modelling. Our results indicate an increase in microviscosity and a decrease in the bending rigidity upon peroxidation of the membranes, contrary to the trend observed with non-oxidized lipids. Fluorescence lifetime imaging microscopy and MD simulations give evidence for the presence of membrane regions of different local order in the oxidized membranes. We hypothesize that oxidation promotes stronger lipid-lipid interactions, which lead to an increase in the lateral heterogeneity within the bilayer and the creation of lipid clusters of higher order.

[1] MSRH, Department of Chemistry, Imperial College London, WoodLane, London W12 0BZ, UK. [2] Center of Physical Sciences and Technology, Saulėtekio av. 3, Vilnius, Lithuania. [3] Departamento de Química en Ciencias Farmacéuticas, Universidad Complutense de Madrid, 28040 Madrid, Spain. [4] Institut Charles Sadron, CNRS and University of Strasbourg, 23 rue du Loess, F-67034 Strasbourg, France. [5] Department of Physics, King's College London, London WC2R 2LS, UK. [6] University of Lyon, ENS-Lyon, CNRS UMR 5182, Chem. Lab., 69342 Lyon, France. [7] University of Lyon, CNRS, INSA Lyon, LaMCoS, UMR5259, 69621 Villeurbanne, France. ✉email: n.brooks@imperial.ac.uk; m.kuimova@imperial.ac.uk

The effect of oxidized lipids on membrane organization is key in controlling the cell cycle and signaling pathways[1–4], and in the development of several pathologies including atherosclerosis[5,6], cancer[7–9], Alzheimer's[10–12], or Parkinson's diseases[2,13]. In addition, enzymatic control of lipid oxidation is a key signaling mediator in metabolic processes, inflammation, immune response, and cell death[14–17]. From a biomedical perspective, lipid oxidation is central to photodynamic therapy (PDT), where targeted photosensitizers produce reactive oxygen species (ROS) to selectively cause apoptosis or necrosis of malignant cells through lipid and protein photo-(per) oxidation[18–20].

The oxidation of lipid tails is often caused by reactions of alkyl chain's double bonds with reactive oxygen species, which can proceed in two ways. In Type II reactions the triplet state of a photosensitizer transfers energy to molecular oxygen to produce its singlet excited state, so called singlet oxygen, $^{1}O_{2}$, which can create lipid hydroperoxide products with a *trans* configuration via an -*ene* reaction mechanism[21,22]. On the other hand, in Type I reactions the triplet state of a photosensitizer participates in electron transfer processes to produce free radicals, and these reactions are expected to yield a chain cleavage, likely at double bond sites[22]. These two processes result in the generation of a highly-polar group and the modification of the chain's architecture within the membrane's core (Fig. 1), and this leads to distinct alterations of the bilayer's biophysical properties.

An ensemble of lipid molecules making up a bilayer can be modelled as a two-dimensional viscoelastic material, where the elastic constitutive parameters (*e.g.* bending rigidity, $\kappa_{c}$, or stretching modulus $K_{A}$) quantify the extent of membrane deformation under a given load and the membrane's microviscosity, $\eta$, the inverse of the membrane's fluidity, determines the lateral diffusion of the membrane-embedded components. Experimentally, it has been suggested that both quantities, $K_{A}$ and $\eta$, are correlated, as they both depend on the intermolecular interaction between the membrane's lipids[23]; yet this relationship is not necessarily universal, and exceptions could be seen in membranes with a more complex structure[24–26].

Existing experimental and computational studies suggest that the less hydrophobic aldehyde and peroxide groups generated as a result of Type I and Type II oxidation can migrate from the

bilayer's hydrophobic core towards the aqueous interface[27–30]. It was reported that this causes an increase in the area per lipid and a decrease in the membrane thickness and the order parameter of lipid tails[27,30–32]. Microscopically, this is manifested in the appearance of an excess membrane area, a lower membrane stretching modulus, $K_{A}$, a higher polarity at the water-lipid boundary[33], a change in membrane morphology[34], an increase in membrane permeability[35] and, for Type I oxidation, in pore formation[36], among others. Strikingly though, all-atom (AA) and coarse-grained (CG) molecular dynamics (MD) simulations predicted decreased diffusion coefficients of lipids[27,37]. Experimentally these computational results were supported by *Borst* et al. who looked at pyrene excimer formation and DPH-HPC fluorescence anisotropy in oxidized lipid mixtures, and reported a decrease in membrane fluidity upon oxidation[38]. In addition, numerical simulations also predict a decrease of the stretching modulus $K_{A}$[27,33] and of the bending rigidity $\kappa_{c}$[27], in agreement with experiments[26] but at odds with the standard correlation between rigidity and microviscosity in simple bilayers.

The membrane's microviscosity is inversely related to the membrane's fluidity. We note that, while viscosity reflects the time-dependent deformation of a substance for a given load, the term microviscosity refers to the molecular mobility of the probe's local environment[39], which in turn defines the in-plane membrane diffusivity. Microviscosity is commonly measured through techniques such as fluorescence correlation spectroscopy[40], fluorescence recovery after photobleaching[41], single particle tracking[42] or time-dependent vesicle deformation[43]. However, these approaches are hard to combine with spatiotemporal mapping of a sample and typically measure diffusion coefficients in a single focal spot, at any one time. Alternatively, environmentally sensitive fluorescent dyes have been used to map the lipid order within the membrane. Previously, microviscosity sensitive dyes[44–47] or polarity sensitive fluorophores[48–51] were extensively used to study membrane properties. It was reported, for example, that lipid peroxidation mediated a depth-dependent increase in membrane polarity[28]. However, to the best of our knowledge, the spatially resolved imaging of lipid organization of the oxidized lipid membranes was not yet reported.

Viscosity-sensitive dyes, termed molecular rotors (MRs), can be used to directly map changes in the bilayer microviscosity upon oxidation. With molecular rotors, the efficiency of the non-radiative decay pathway is coupled to the molecule's intramolecular rotation. In less crowded or low viscosity environments intramolecular motion is not restricted and, therefore, the non-radiative decay dominates, leading to a decrease in the fluorescence quantum yield and lifetime. Using the lifetime to quantify the environment's microviscosity is particularly advantageous, as it is independent of a probe's concentration, and is largely unaffected by optical properties of the medium and the excitation setup. This advantage has been exploited to quantify the micromechanical properties of model, prokaryotic and eukaryotic lipid bilayers through Fluorescence Lifetime Imaging Microscopy (FLIM)[52–56]. Furthermore, molecular rotors were used to study membrane's behavior under osmotic pressure[25], flow-induced shear[57], drug treatment[58], hyper-gravity conditions[59] or oxidative stress[53]. In the latter case, we previously observed an increase in membrane microviscosity following photo-induced Type II oxidation of unsaturated lipid bilayers. However, the decrease in membrane fluidity contrasted with the expected disruption of the hydrophobic core by the hydroperoxides.

In this work, we elucidate the mechanism responsible for the increase in microviscosity of oxidized membranes and its uncoupling from the bilayer's elastic properties. Molecular rotors directly visualize domain formation in single component oxidized

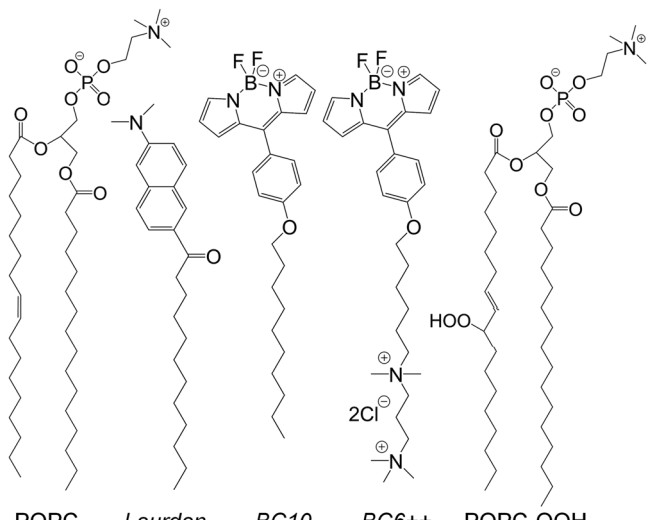

**Fig. 1 Schematic of the lipid and dye structures used in this work.** Peroxidation of POPC results in the addition of a peroxide group (-OOH) which locates at either 9' or 10' position (see ESI). Membrane order is measured using the dyes Laurdan, BC10 and BC6 + + .

lipid bilayers via FLIM, while MD simulations indicate that such behavior is driven by attractive interactions between the lipid peroxides. Our results provide a theoretical framework to understand the drastic effects of lipid peroxidation on the biophysical properties and lateral organization of lipid membranes, which could be of use in the design of oxidation-based therapies[8,60] and in the engineering of synthetic cells[61].

## Results and discussion

**Effect of lipid peroxidation on membrane structure.** Initially, we studied the effect of lipid peroxidation on 1-palmitoyl-2-oleoyl-sn-glycero-3-phosphocholine (POPC) bilayers by combining the microviscosity data obtained using two well-characterized BODIPY-based molecular rotors[45,55] (BC10 and BC6++, Fig. 1) and the Laurdan polarity-based fluorescent probe, with the data from X-Ray diffraction.

We have previously used BODIPY-based molecular rotors to measure a large increase in microviscosity upon Type II photooxidation (during PDT) of unsaturated lipid 1,2-dioleoyl-sn-glycero-3-phosphocholine (DOPC), from *ca* 180 cP to 480 cP[53]. We also measured a large microviscosity increase, from *ca* 80 cP to 350 cP, in hydrophobic organelles of cultured mammalian cells exposed to PDT with Type II photosensitisers[62,63]. Hence, we aimed to directly relate the observed increase in microviscosity with an increased presence of peroxidized lipid.

Consequently, we synthesized POPC-OOH and prepared mixed liposomes containing mixtures of POPC/POPC-OOH with increasing fractions of the oxidized lipid. As shown in Fig. 2a, b, BC10 lifetime increased from $1.8 \pm 0.1$ to $2.1 \pm 0.07$ ns, indicating a change in microviscosity from $159 \pm 21$ cP for pure POPC bilayers to $241 \pm 16$ cP for pure POPC-OOH membranes, according to the previously published lifetime-viscosity calibration[64]. We note that molecular rotor BC10 is localized in the hydrophobic core of the membrane;[55] as the rotor shows a polarity and temperature-independent responses within the studied viscosity range[65], it is expected that changes in BC10 lifetime are truly representative of changes in the bilayer's microviscosity. This was further supported by the MD simulations described later and by FRAP experiments on pure POPC and POPC-OOH membranes (Fig. S1). The change in microviscosity reported here is significantly higher than that reported in[26], where interfacially localized TMA-DPH was used as a probe, which was affected by both hydration and viscosity.

Similar results were obtained when BC10 labelled POPC giant unilamellar vesicles (GUVs) were subjected to in situ Type II oxidation (Fig. 3) using singlet oxygen production from the photosensitiser tetraphenylporphyrin (TTP), Fig. S2. TPP is a hydrophobic porphyrin which localizes inside the lipid bilayer and produces significant amounts of singlet oxygen upon irradiation[53,66]. The photo-induced increase in microviscosity (from $121 \pm 11$ cP to $298 \pm 14$ cP, Fig. 3d) is comparable to that observed for premixed synthetic lipids, Fig. 2, indicating that, indeed, lipid-hydroperoxide formation *via* reactions with singlet oxygen is responsible for the significant increase in membrane microviscosity, as a result of Type II peroxidation.

Of note, the microviscosity values detected for the oxidized membranes are comparable to those obtained from the liquid-ordered region of phase-separated vesicles, ~290 cP, and are significantly higher than those recorded for liquid disordered phase lipids, ~130 cP[55]. Similar to the highly ordered lipid "rafts", these high microviscosity values reinforce the suggested possible roles of lipid oxidation products in signal transduction[67,68].

The higher order of POPC-OOH containing membranes was also confirmed by monitoring Laurdan fluorescence. Unlike the BC10 molecular rotor, Laurdan is sensitive to the polarity of the surrounding solvent and is capable of reporting on both the

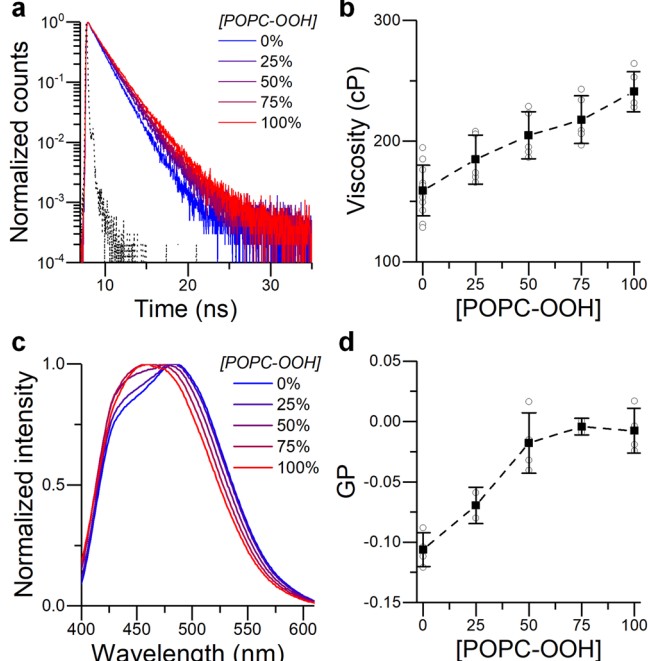

**Fig. 2 Spectroscopic characterization of POPC/POPC-OOH large unilamellar vesicles (LUVs). a** Time resolved decay traces and (**b**) calculated microviscosity of BC10 labelled LUVs. **c** Emission spectra and (**d**) calculated GP of Laurdan-labelled LUVs. Instrument response function (IRF) is shown in black (**a**). Data shown as mean ± S.D. ($n \geq 3$ independent repeats).

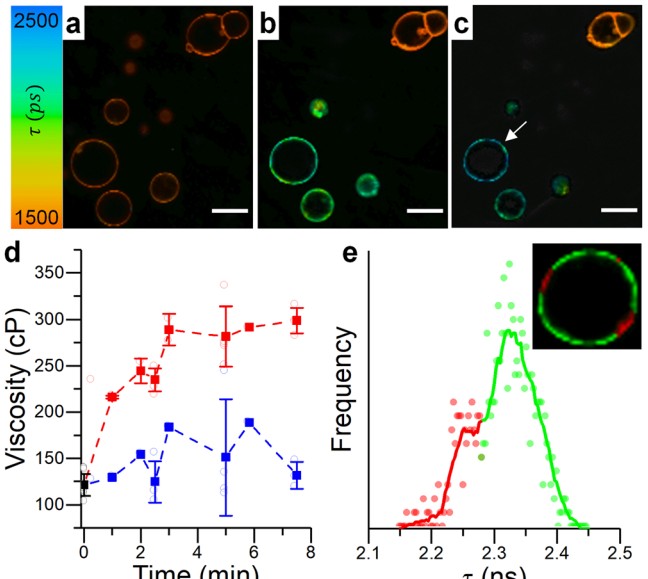

**Fig. 3 Type II photo-oxidation of POPC GUVs, monitored by FLIM of BC10. a–c** FLIM images recorded at 0 s, 150 s and 450 s. Only the bottom left section was irradiated. Scalebar: 20 μm. **d** Changes in microviscosity recorded for irradiated (red) and non-irradiated (blue) GUVs. Data shown as mean ± S.D ($N \geq 3$ GUVs in $n = 3$ independent repeats). **e** Lifetime distribution histogram and a zoomed in FLIM image of the irradiated GUV indicated by the white arrow in (**c**). Discrete color-coding of the inset FLIM image has been used to highlight areas of different microviscosity.

membrane's polarity and hydration[69,70]. The commonly reported measure of polarity is the general polarization (GP) value (defined in Methods), which is assumed to be proportional to lipid packing density for simple membrane compositions[23]. As seen in Fig. 2c, d, increasing amounts of POPC-OOH changed Laurdan's GP from $-0.11 \pm 0.01$ for pure POPC membranes to $-0.01 \pm 0.02$ for fully peroxidized bilayers, consistent with a decrease in polarity and hydration and an increase in lipid packing as a result of peroxidation. This data is consistent with a previously reported trend[26] and with the increase in membrane microviscosity reported by using molecular rotor BC10. We note that the relation between Laurdan's GP and microviscosity does not appear to be significantly affected by the presence of lipid peroxides compared to standard temperature-induced changes in the membrane's structure (Fig. S3a) nor does the presence of charged lipids or hydrogen-bonding affect the readout from the BC10 probe (Fig. S3b, Fig. S17).

In contrast, our previous work using 3-hydroxyflavone based reporters[28] suggested that the local polarity increased in a depth-dependent manner, upon increasing the fraction of POPC-OOH. This should, in principle, cause a decrease in Laurdan's GP, opposite to what was observed here. To attempt to understand this discrepancy, we measured Laurdan's emission spectra in vesicles composed of POPC with an increasing fraction of POPC-OOH, at increasing temperature. We hypothesized that a higher temperature may favor the migration of the -OOH groups towards the surface, due to looser lipid packing. Each spectrum was fitted using 3 lognormal curves, to represent three contributions: apolar (centred at ~425 nm), polar aprotic (~450–470 nm) and polar protic (~480–510 nm) components, according to a published procedure[69].

These peaks were assigned to Laurdan localized at three different heights within the membrane, reporting on the hydrocarbon region, membrane's polarity and membrane's hydration, respectively[69]. The fitting results are shown in Fig. S4. As shown in Fig. S5b, the Laurdan's GP value in membranes containing 50 and 100% POPC-OOH was higher than that obtained in pure POPC bilayers, regardless of the temperature. However, when POPC-OOH was present, the contribution from membrane polarity to Laurdan's spectra increased with temperature whereas the contribution from the membrane hydration component did not vary significantly (Fig. S4). This is consistent with an increased migration of hydroxyl groups towards the membrane-water interface, increasing the polarity at the phosphate level, in agreement with existing reports[7,26–28,30,33]. Therefore, given the most red-shifted component (centered at ~480–510 nm) does not vary significantly, an increase in the polarity contribution should be reflected in a larger GP, in agreement with our observations.

Overall, these results contribute evidence that peroxidized lipids increase the membrane's polarity. However, an unexpected result is that the membrane's microviscosity was also increased. We tested whether this increase could be due to the *trans* configuration adopted by lipid peroxides[21]. We measured BC10 lifetime in membranes composed of POPC and the *trans* lipid DEPC (1,2-dielaidoyl-sn-glycero-3-phosphocholine, the *trans* isomer of DOPC). As shown in Fig. S6, a slight increase in lifetime (and microviscosity), from $159 \pm 21$ cP to $191 \pm 19$ cP, was observed in the case of LUVs composed of pure DEPC. Notably, this increase is significantly lower than for POPC/POPC-OOH containing membranes. This suggests that the *trans*-nature of oxidized POPC is not solely responsible for the observed increase in microviscosity and there is a more complex cause for the decrease in membrane fluidity upon peroxidation.

To further investigate this phenomenon, we performed small- and wide-angle X-ray diffraction (SAXS/WAXS) on POPC bilayer

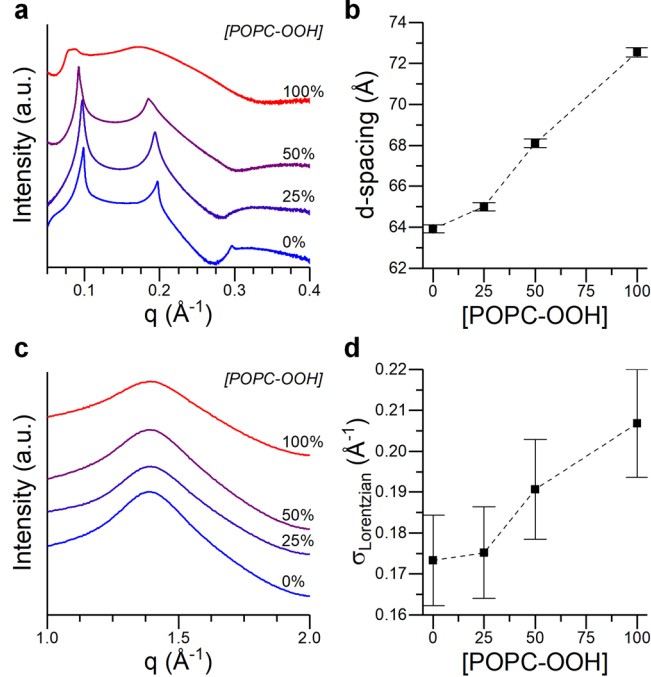

**Fig. 4 X-Ray scattering intensity profiles for POPC/POPC-OOH bilayers.** **a**, **b** SAXS profile and interlamellar spacing calculated from the 1st Bragg peak. **c**, **d** WAXS traces and width of the fitted Lorentzian component. Data shown as mean ± S.D. S.D. in POPC-OOH containing membranes was estimated according to the S.D. of pure POPC samples, see ESI for details.

stacks containing an increased fraction of either POPC-OOH or DEPC. The position of the SAXS peaks is determined by both the actual bilayer thickness and the water layer in between the lamellae, both affected by the presence of lipid peroxides[30]. Our results, shown in Fig. 4a, b, indicate a swelling of the lipid stacks by ~9 Å when POPC-OOH completely replaces POPC, in agreement with previous reports[26].

These samples were prepared with excess hydration, allowing the water to move into and out of the interlamellar space. As these membranes are formed from neutral lipids, changes in the interlamellar distance would be determined by the balance between the Helfrich repulsion forces (due to membrane's undulations) and the dispersive attractive forces. Under these conditions, an increase of the repeat distance can be viewed as the result of an increase in the repulsion strength, compatible with the expected reduction of the bending modulus of the bilayer. This reduction of the bending modulus is further supported by the observed broadening of the SAXS peaks, and the intensity decrease of the higher-order peaks at higher POPC-OOH fractions. According to Caillé theory[71], the peak broadening corresponds to a larger stack fluctuation and lower bending modulus, the later also confirmed by flickering spectroscopy (Fig. S7). This result matches previous observations[26], and is consistent with the decrease of the phase transition temperature ($T_m$) observed by DSC (Fig. S8), which suggests the presence of POPC-OOH could lower the cohesive forces between POPC molecules. On the contrary, exchanging POPC-OOH for DEPC resulted in a significantly lower increase in d-spacing, compatible with the expected increase in membrane thickness corresponding to a *cis-trans* isomerization. Furthermore, the use of DEPC did not result in peak broadening, indicating membrane elasticity was not compromised (Fig. S9).

On the other hand, the position and shape of the WAXS peak report on the lipid packing density (i.e., area per lipid). When POPC was substituted by POPC-OOH the WAXS peak position,

corresponding to the fluid lamellar phase (d ~ 0.45 nm, area per lipid ~68 Å²), did not shift; suggesting the average inter-lipid spacing remained unchanged (Fig. 4c, d). However, the WAXS peaks became significantly broader, predominantly towards larger q-values (corresponding to lower lipid spacing), which could be compatible with a wider area per lipid (APL) distribution, as would be expected in a bilayer displaying lateral heterogeneity. In contrast, for DEPC containing bilayers, the WAXS peak shift was small, which agreed with the small changes observed in BC10 lifetime and SAXS traces, while no broadening of the WAXS spectrum was evident. This indicates a gradual, homogeneous thickening of the membranes in case of the DEPC addition, with lipid molecules getting closer together and similar lateral heterogeneity.

All in all, these results suggest that, even if the overall lattice parameter remains the same, there is an increase of heterogeneity in the lateral distribution of POPC-OOH containing membranes; a behavior that could be compatible with lipid bilayers displaying membrane regions with distinct degrees of lipid packing. This is further supported by the concurrent increase in microviscosity and decrease in bending rigidity, a deviation from the generalist trend where both magnitudes are directly correlated[23]. A similar effect, where lipid bilayers follow a non-classical behavior, has been reported for membranes containing highly ordered lipid clusters of low fluidity[24,25].

**Peroxide-induced domain formation.** Lipid peroxide-induced phase separation was previously reported in the literature[22,72,73], where the presence of lipid hydroperoxides was thought to increase the membrane's tension, leading to the emergence and/or fusion of highly ordered lipid domains in vesicles containing a mixture of saturated and unsaturated lipids, and cholesterol[74]. However, to the best of our knowledge, the presence of lipid domains in membranes composed of a single lipid together with its hydroperoxidised analogue has not yet been described.

In fact, probing such a phase separation using conventional microscopy techniques (i.e., using fluorescently tagged lipids that partition into a given phase) is challenging, as the two domains are expected to have a relatively similar packing (see for example WAXS data in Fig. 4). To overcome this issue, we mapped the microviscosity of POPC GUVs with increasing fractions of POPC-OOH via FLIM using viscosity sensitive molecular rotors.

Here, we substituted BC10 for the well characterized water-soluble molecular rotor BC6++, because of the apparent interaction of BC10 with oxidized lipids during the electroformation process (Fig. S10, see ESI for further discussion). We confirmed that the change in GUVs microviscosity at increasing fractions of peroxidized lipid from 144 ± 6 cP to 239 ± 16 cP, recorded using BC6++, (Fig. 5) was comparable to the changes seen in LUVs with BC10, Fig. 2. We verified that there is also a small difference in microviscosity values seen in LUVs stained with BC10 and BC6++ (Fig. S11).

Notably, by performing FLIM microscopy, we were able to detect a significantly wider microviscosity distribution of POPC-OOH containing GUVs, compared to pure POPC (Fig. 5). The wider lifetime histograms as well as a non-gaussian histogram shape was seen for both 50 and 100% POPC-OOH (Fig. S12) gave evidence for the presence of membrane regions with significantly higher order, Fig. 5b-d. The increase in membrane heterogeneity was also seen in POPC GUVs that were photooxidised in situ (Fig. 3e and Fig. S13). We interpret this data as evidence for domain formation in POPC/POPC-OOH GUVs. This conclusion is consistent with the increased width of the WAXS peaks.

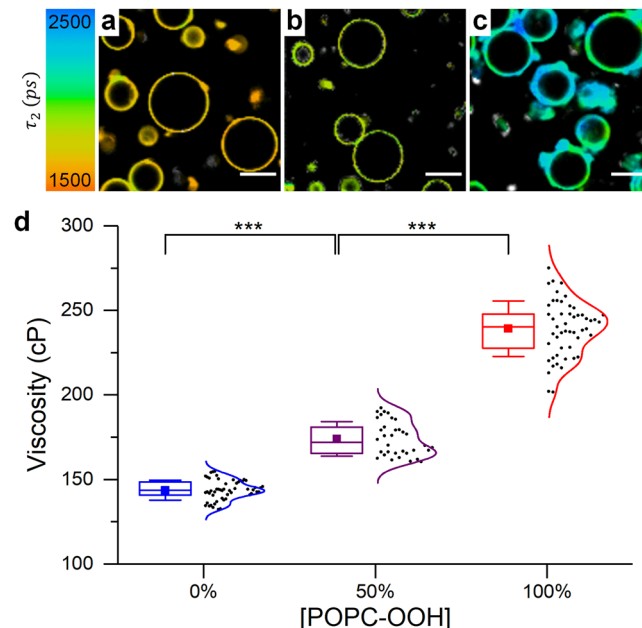

**Fig. 5 FLIM imaging of POPC/POPC-OOH GUVs. a–c** Sample FLIM images of BC6++ stained GUVs with increasing fractions of POPC-OOH. Scale bar: 20 μm. **d** Calculated membrane microviscosity using the reported rotor calibration[85]. Box plots display the 25–75% range, error bars represent ± S.D., median is shown by a horizontal line and mean by a dot of $N \geq 30$ GUVs from $n = 3$ independent repeats.

To further explore evidence for the existence of lipid domains, we measured changes in Laurdan GP and BC10 intensity of POPC-OOH containing LUVs at increasing temperatures (Fig. S5). We expected these temperature-dependent curves to be linear if only one lipid phase is present. However, it could be seen that in the presence of POPC-OOH, a clear change in the slope of both GP and fluorescence intensity vs T, for both Laurdan and BC10 signals, was observed. The presence of these points of changing gradient at ~50 °C gives further evidence for the phase-separation, with transition temperature consistent between both methods. A similar behavior was also seen upon heating POPC-OOH GUVs at 60 °C (Fig. S14).

Our data suggests, for the first time, that lipid peroxides can cluster together creating regions of higher order, even in the absence of canonical $L_o$ domain-forming lipids. Prior electrical current recordings of POPC-OOH containing membranes by Corvalán et al.[36] also indirectly suggested that lipid peroxides could cluster together, leading to the formation of ion lipid conductive pores. Altogether, these findings indicate that the presence of lipid peroxidation products can promote the appearance of lipid clusters of higher local order, which can play a crucial role in signal transduction and mechanical behavior of the cell's membrane.

*MD simulations of peroxidized membranes.* In order to further interpret our experimental observations, we studied the atomic organization of POPC-OOH containing membranes by using all-atom (AA) molecular dynamics (MD) simulations.

Figure 6d–f shows the electron density profile (EDP) corresponding to the lipid species, water and BC10 as a function of the z-coordinate (normal to the lipid bilayer interface) from the AA-MD simulations. An increased fraction of POPC-OOH (0, 50 and 100%mol) leads to a smoother EDP trace and to a larger lipid density at the membrane's midplane. This suggests a higher degree of delocalization of the lipid molecules (Fig. 6a–c), in

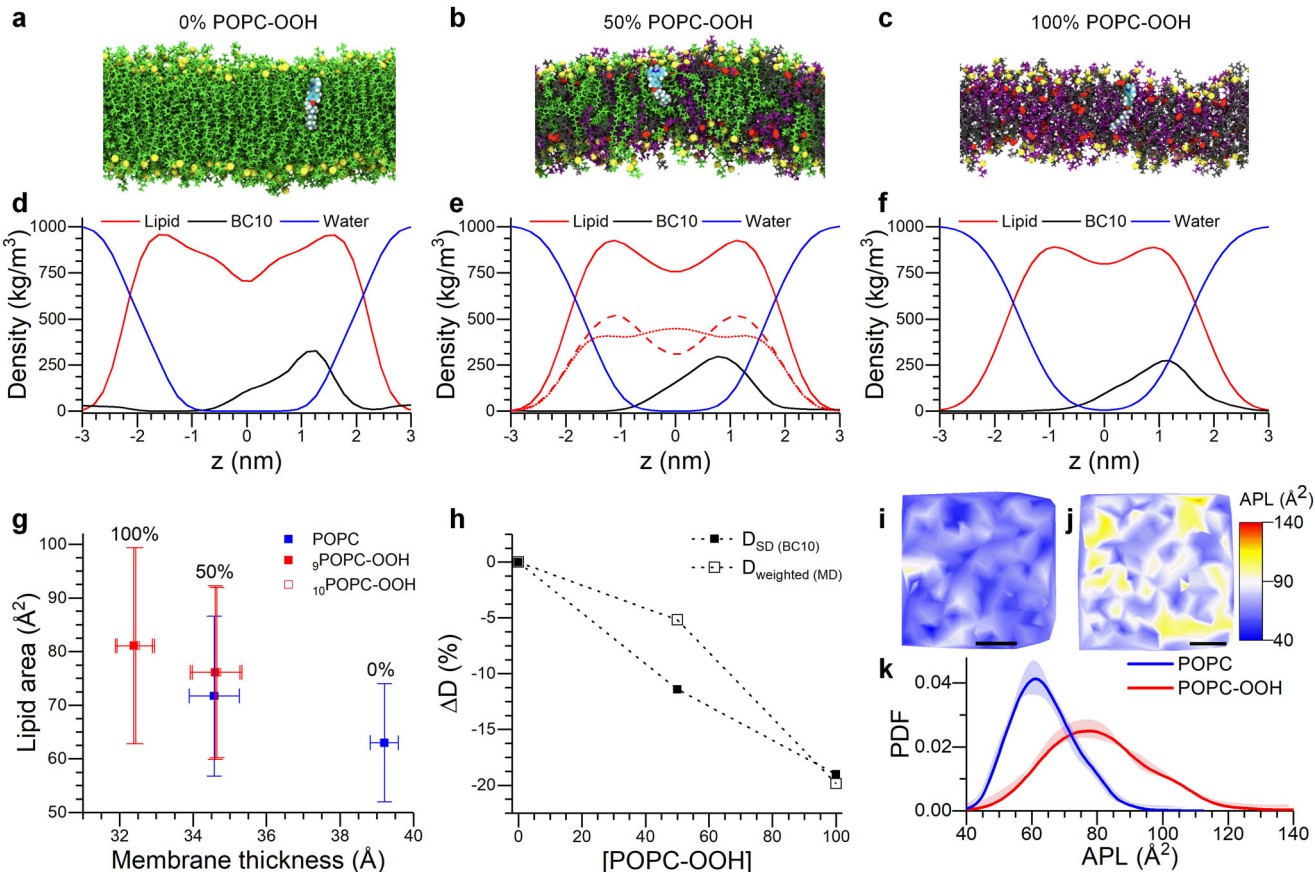

**Fig. 6 MD simulations of POPC-OOH containing membranes. a–c** Lateral snapshots of the simulated bilayers containing increasing amounts of POPC-OOH. Colour coding: Green-POPC, Gray-oxPOPC9, Purple-oxPOPC10 (see Fig. S15 for details), Yellow-Phosphorous, Red-Oxygen. BC10 is also shown (cyan), within the leaflet regardless of the membrane's composition, but closer to the phospholipid headgroups in the 100% POPC-OOH membrane. **d, f** Electron density profiles of membranes containing increasing amounts of POPC-OOH. In (**e**) the red traces correspond to all lipids (continuous trace), POPC (dotted) and POPC-OOH (dashed). **g** Area-thickness relationship for lipid peroxide containing bilayers. Data shown as mean ± S.D. from two independent simulations. **h** Relative variation of the 2-dimensional diffusion coefficient obtained from the MD simulations (□) and from molecular rotor data using the Saffman-Delbrück equation (■). **i, j** Snapshots of a coarse-grained simulation showing the area per lipid (APL) maps of pure POPC (**i**) and POPC-OOH (**j**) membranes. Scalebar: 30 nm. **k** Bimodal APL distribution seen for the oxidized membranes (**j**), supporting the observation of lipid segregation in POPC-OOH membranes.

agreement with the disruptive effect caused by the snorkelling of ~40% of the oxidized chains towards the membrane interface (Fig. S15). This effect is reflected in the change of the POPC EDP for pure and mixed (50% POPC-OOH) peroxidized bilayers (Fig. 6e, f). Here, the snorkelling POPC-OOH chains lead to POPC depletion at the water/lipid interface and, simultaneously, to a higher density of the lipid tails from the non-oxidized POPC lipid molecules and the bilayer's core. We also observed an increase in water penetration with increasing amounts of POPC-OOH, in agreement with previous reports[36,37,75]. With regard to the location of BC10, peak density was within the highest density region of the membrane and in a similar location to our previous studies[55,76,77], regardless of the bilayer's composition (Fig. S16). In addition, we observed no significant interaction between BC10 and the oxidized moiety (Fig. S17). Altogether, this data gives us confidence that the change in BODIPY rotor's lifetime signal does, indeed, reflect the changes in membrane's packing, rather than a relocalization of the sensor during peroxidation.

Regarding the bilayer's structural properties (Fig. 6g), we calculated a decrease in the bilayer thickness (determined as the phosphate-phosphate distance) from 39.2 ± 0.4 Å for pure POPC to 32.4 ± 0.6 Å for pure POPC-OOH membranes. This decrease in membrane thickness was accompanied by an increase of the mean area per lipid from 63 ± 10 Å² to 81 ± 16 Å², in agreement

with previous reports[7,30]. For the mixed membrane composition (50% POPC-OOH) the area per lipid of POPC molecules increased to 72 ± 14 Å² while that of POPC-OOH decreased to 76 ± 16 Å².

We also measured the conformational flexibility of the acyl chains through the order parameter $S_{cd}$ given by:

$$S_{cd} = \frac{1}{2}\left\langle 3\cos^2\theta - 1\right\rangle \tag{1}$$

where $\theta$ is the angle between the bilayer normal and the carbon-hydrogen vector of a carbon atom in an acyl tail, and the average is taken over time and over all molecules of a given species within the membrane. As depicted in Fig. S18, the fully saturated sn1 chain was generally less ordered for POPC-OOH lipids, in agreement with the observed increase in lipid area for peroxidized bilayers. In the case of the sn2 chain, however, the effect of the hydroxyl group in POPC-OOH molecules becomes dependent on the carbon number and is characterized by a sudden increase of $S_{cd}$ at the carbon positions 8 and 11, which are attributed to the snorkeling of the peroxidized chain towards the membrane interface. It is likely that the looping of the peroxidized sn2 chain and the increase in ordering would increase the local lipid areal density, thereby increasing the membrane's microviscosity, as directly detected by FLIM of molecular rotors in this work.

To test this hypothesis, we calculated the lateral mean squared displacement (MSD) of the different lipid species and used it to extract the lateral diffusion coefficient D (see ESI for details). When comparing POPC-OOH and POPC lipids, we observed a decrease of ~20% in the diffusion coefficient in the case of POPC-OOH (Fig. 6h). This is in agreement with the values of diffusion coefficients derived using the Saffman-Delbrück formula from the microviscosity data reported by molecular rotor BC10. For mixed membranes of equimolar POPC/POPC-OOH composition we observed some discrepancy between the calculated values of D and measured viscosities, which may be due to lipid cooperativity and clustering.

Finally, to assess the degree of lipid mixing within membranes containing oxidized species, we defined the fractional enrichment of species $i$ around molecules $j$, $E_{ij}$, within a 12 Å neighbourhood as:

$$E_{ij} = \frac{C_{j,local}}{C_{j,bulk}} \tag{2}$$

where $C_{j,local}$ and $C_{j,bulk}$ are the local and bulk concentrations of species $j$ around species $i$; hence a higher value of $E_{ij}$ will indicate a preference of molecule types $i$ and $j$ to cluster together. Our results suggest that the oxidised species prefer to associate with other oxidised lipid molecules, ($E_{ij} > 1$, $i = j$) while POPC lipids had no association preference ($E_{ij} < 1$, $i \neq j$). This result indicates the formation of lipid clusters rich in POPC-OOH, in good agreement with our experimental data and previous predictions[36]. In order to visualize these results, we created a map showing the area per lipid (APL) and the distance between the C9 chain position (for pure POPC) or the -OOH group (for pure POPC-OOH) to the membrane's midplane. As depicted in Fig. 6i and Fig. S19, significant spatial heterogeneity was observed in oxidized membranes compared to their non-oxidized counterparts. This result confirms the presence of phase-separated lipid domains, which we experimentally observed using molecular rotors, even in single component oxidized bilayers. To the best of our knowledge, this is the first instance where lipid clustering has been reported in otherwise single-component membranes.

We note that, although the presence of lipid clusters in single-component membranes is apparently inconsistent with the Gibbs phase rule, it must be considered that POPC-OOH is found in two states (e.g., with the -OOH group either embedded within the hydrocarbon region or snorkelling towards the membrane's surface). While, thermodynamically, this should not allow formation of two phases, we believe that interconversion between the snorkelling and non-snorkelling configurations is slow, leading to kinetic trapping of the molecular states, which allows formation of the observed membrane heterogeneity. This is supported by our MD results that suggest approximately 40% of the POPC-OOH molecules display a snorkelling behavior during the simulation time (Fig. S15). In addition, apparent violations of the Gibbs rule are commonly seen in lipid vesicles, although poorly understood. For example, pure DPPC membranes display gel-fluid domain coexistence over a temperature range around the main transition, which has been attributed to finite-size effects and metastable states[78].

To gain a deeper understanding of the molecular interactions driving this lipid segregation, we created contact maps between the different lipids, calculated by all-atom simulations. As shown in Fig. S20, a significant amount of contact is seen between the oxygens in the oxidized -OOH tail of the lipids (O9 & O10) and also between those oxygens and the oxygens in the ester groups of lipid heads (O6 & O7). These contacts are indicative of hydrogen bonds formed between the OH group in the neighboring oxidized tails, as well as between the oxygens in the oxidized tails and ester groups of neighboring lipids. These bonds could serve as a mechanism for the local clustering of the oxidized species, which were observed both in our all-atom MD, and, over longer time scales, the formation of domains observed in our experimental FLIM investigations measuring lipid packing using molecular rotors.

By combining advanced microscopy and environmentally sensitive probes, X-ray diffraction experiments and MD simulations, we show that lipid peroxidation results in an increased interaction between the lipids. Snorkelling of the -OOH containing alkyl chains towards the membrane surface will increase the area per lipid, but will also increase the molecular crowding within the bilayer and will promote H-bonding interaction between peroxidized lipids, leading to a decreased diffusivity and membrane fluidity. In addition, the emerging lipid-lipid interactions would stabilize the formation of lipid regions of higher microviscosity. Simultaneously, snorkelling of the peroxidized chains would decrease the chain-chain interactions at the bilayer's midplane, causing a decrease in membrane thickness and bending rigidity. Overall, this is reflected in the disruption of the canonical correlation between membrane elasticity and viscosity and the emergence of more-ordered lipid regions in otherwise single-component membranes (Fig. 7). The change in the relationship between membrane viscosity and bending rigidity suggests a shift in the balance between the inter-leaflet and intra-leaflet lipid interactions; hence, we anticipate the drastic effect of lipid peroxidation on the membrane's structure will be key in the cellular response to environmental stress. In addition, we envision the proposed membrane-remodeling mechanism could be exploited in the design of membrane-sorting artificial cell systems.

## Methods

**Materials.** Lipids solutions in CHCl₃ of 1-palmitoyl-2-oleoyl-sn-glycero-3-phosphocholine (POPC), 1,2-dielaidoyl-sn-glycero-3-phosphocholine (DEPC) and 1-palmitoyl-2-azelaoyl-sn-glycero-3-phosphocholine (PAzePC) were purchased from Avanti Polar Lipids® and diluted to a 20 mM stock before use. The BODIPY-based

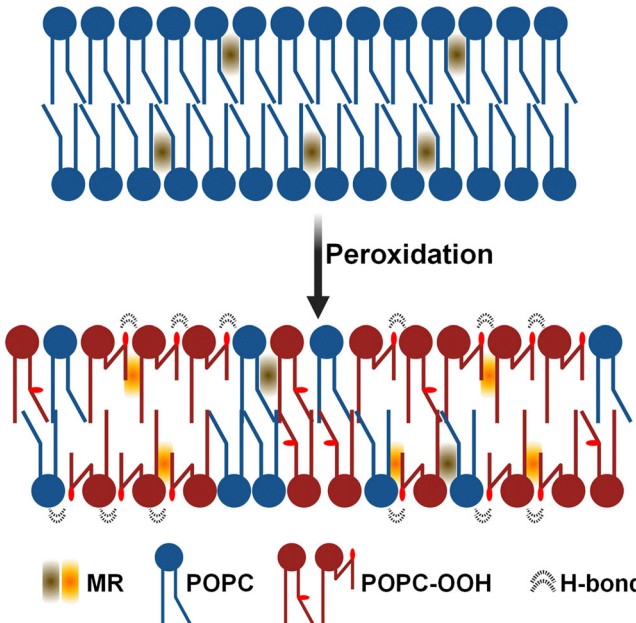

**Fig. 7 Effect of lipid peroxidation on membrane architecture.** Snorkelling of the lipid tails leads to an increased lipid packing towards the interface, yet this causes the increase of the individual area per lipid. Concurrently, chain snorkelling creates a "void" at the membrane's midplane, which lead to a lower membrane thickness and elastic properties.

dyes BC10 and BC6 + + were synthesized according to the previously published procedures[55]. Stock solutions were prepared in CHCl₃ (300 μM for BC10 and Laurdan) or DMSO (3 mM, BC6 + +). 99.8% deuterated methanol (MeOD) for NMR analysis was supplied by Fisher/Acros Organics. All other reagents were purchased from Sigma Aldrich® or VWR and used without further purification. Solvents for fluorescence studies were of spectrophotometric grade.

**Lipid hydroperoxidation.** Hydroperoxidized POPC (POPC-OOH) was obtained following a photochemical technique using methylene blue (MB) as a photosensitizer[28,79]. Briefly, to 6 mL of POPC in MeOD (5 mg mL⁻¹), 50 μl of a 6 mM MB solution in MeOD was added, to reach a concentration in MB of 50 μM (20.8 μg mL⁻¹). The obtained solution was illuminated under a red light (λ = 656 nm), and kept under constant oxygen flux and stirring for 10 min. The solution was then characterized by NMR, to determine the hydroperoxidation degree (see Fig. S21 for typical NMR spectra). Lipids were purified by dialysis and absence of lipid degradation was checked by a last NMR spectrum acquisition. Each final hydroperoxidized lipid solution was dried and redispersed in MeOD, to reach a 10 mg mL⁻¹ concentration. The POPC-OOH solutions in MeOD were kept at −20 °C until further use.

**Large Unilamellar Vesicle (LUV) formation.** POPC and POPC-OOH stock solutions were mixed at the appropriate molar ratio and, where appropriate, Laurdan or BC10 were added at a 1:200 dye:lipid molar ratio. CHCl₃ was then removed using a rotatory evaporator to create a lipid film. The dried lipids were then hydrated to a final concentration of 1 mM lipid using a 400 mM sucrose solution and the vial was vortexed until the solution turned cloudy, after which it was extruded 21 times through a 200 nm polycarbonate filter (Avanti Polar Lipids®).

For BC6 + + stained LUVs, pure liposomes were prepared as described above, without adding the dye before a film formation. After extrusion, the liposome solution was incubated for ~1 h with a 10 μM BC6 + + solution to stain the bilayer (~1:150 dye:lipid molar ratio, final DMSO concentration <1%v/v).

LUVs were resuspended in 400 mM glucose to a 0.1 mM lipid concentration before measurement. Vesicle size for POPC-OOH containing membranes was confirmed to be in the expected range (~190 nm diameter) using Dynamic Light Scattering (Malvern Panalytical, Zetasizer Ultra).

**Giant Unilamellar Vesicle (GUV) formation.** 30 μL of a 1 mg mL⁻¹ lipid solution with the indicated POPC/POPC-OOH molar ratio, supplemented with 0.05% BC10 when required, was spread onto an ITO slide. After drying for >1 h under vacuum, a PDMS spacer was pressed onto the slide, the chamber was filled with a 400 mM sucrose solution and was then closed with a second ITO slide. The electroformation protocol consisted of the application of 1V_pp@10 Hz electric field for 90 min, followed by a 30 min detachment phase at 1V_pp@2 Hz. We note that the low voltage was used to avoid lipid oxidation. For BC6 + + labelled GUVs, vesicles were incubated for ~1 h with a 10 μM BC6 + + solution to (<1%v/v DMSO). GUVs were then diluted ten-fold before imaging (using a custom-made PDMS chamber on top of a BSA-coated glass slide). Samples where the microviscosity for pure POPC GUVs was significantly different to LUVs of the same composition were discarded due to suspected electroformation-induced oxidation.

**Fluorescence spectra and lifetime measurements.** Samples were placed in low volume 500 μL cuvettes (10 mm path length). Fluorescence wavelength-corrected emission spectra was recorded using a Horiba Yvon Fluoromax 4 fluorimeter under 404 nm (BC10) or 360 nm (Laurdan) excitation. The time-resolved fluorescence decay traces were obtained using a Horiba Jobin Yvon IBH5000 F time-correlated single photon counting (TCSPC) instrument. The sample was excited using a 404 nm pulsed laser (NanoLED) and the decay trace was recorded at 515 nm, until peak counts reached 10.000. These traces were then fitted to either a mono-exponential (BC10) or biexponential (BC6 + +)[55] decay models using DAS® software. For the later, the longer lifetime component was used to calculate the membrane's microviscosity[55]. Temperature was controlled through a Peltier cell (fluorimeter experiments, precision ±0.5 °C) or a water bath (TCSPC experiments, precision: ±1 °C), and was set to 22.5 °C unless stated otherwise.

**Fluorescence Lifetime Imaging Microscopy (FLIM).** GUVs were diluted 10-fold in 400 mM glucose solution and added to a BSA-coated observation chamber. Lifetime images were obtained using a Leica TSC SP5 II inverted confocal microscope and a 20x air objective. A Ti:Sapphire laser (Coherent, Chameleon Vision II, 80 MHz) provided two-photon excitation at either 930 nm (premixed vesicles) or at 900 nm (for the in situ oxidation experiments) and fluorescence emission was collected between 500–580 nm (premixed vesicles) or between 500 and 600 nm (for in situ oxidation). FLIM images (256 × 256 pixels, 256 channels) were acquired using a TCSPC card (Becker & Hickl GmbH®, SPC-830). Instrument response function (IRF) was measured using the second harmonic generation signal from urea crystals. The lifetimes were calculated by fitting the decays to either a monoexponential (BC10, minimum 200 counts per pixel at peak after binning) or biexponential (BC6 + +, minimum 500 counts per pixel at peak after binning) models.

**X-Ray diffraction experiments.** Dry samples of a given lipid mixture (20 mg total mass) were hydrated with DI water to 70% and subjected to 15 freeze-thaw cycles to ensure a proper lipid mixing. The sample was then loaded into a 2 mm diameter polymer capillary tube and sealed with a rubber stopper. SAXS and WAXS measurements were performed at beamline I22 (Diamond Light Source, UK)[80]. Analysis of the diffraction patterns is described in the ESI.

**All-atom (AA) simulations.** All atom simulations of three systems have been performed: one consisting of 100% POPC, one consisting of 50% POPC and 50% POPC-OOH (25% oxPOPC9 & 25% oxPOPC10) and one consisting of 100% POPC-OOH (50% oxPOPC9 & 50% oxPOPC10). Each system contains a lipid bilayer consisting of 200 lipids per leaflet surrounded by an aqueous environment, which contains approximately 50 water molecules per lipid. Packmol was used to build each of the three membrane systems[81].

The lipids were modelled using the CHARMM36 forcefield. The standard parameters were used for POPC, oxPOPC9 and oxPOPC10. The models of the oxidized PC lipids were then modified to incorporate the forcefield terms of the oxidized group as reported by Wong-Ekkabut et al.[82]. Meanwhile the water molecules were modelled using the CHARMM-modified version of the TIP3P potential[83]. Finally, the BODIPY molecule was modelled with the same forcefield as we have used previously[76,77].

Each lipid membrane system was minimized and then equilibrated to a temperature of 300 K and a pressure of 1 bar following the simulation protocol prescribed by CHARMM-GUI[84]. Then the membranes are simulated for 500 ns at 300 K and 1 bar in order to make sure the membranes are thoroughly equilibrated before the BODIPY dye molecules are added to the systems. Then the BODIPY molecules are inserted into the final configuration of these long simulations. One BODIPY molecule is added to each system into the aqueous phase of the bilayer systems. Then the systems are simulated for a further 200 ns at 300 K and 1 bar. The temperature was controlled by a Nosé-Hoover thermostat and a Parrinello-Rahman barostat was used to control the pressure. Three replicas were created for each of the bilayer systems with BODIPY and simulations were carried out for each. The results presented in this manuscript are taken by performing statistical analysis of the results from each of the three replicas for each system.

**Supplementary methods.** For other methods (NMR characterization of POPC-OOH, estimation of the diffusion coefficient from BC10 lifetime, FRAP, Laurdan GP and spectral decomposition analysis, analysis of intra-vesicle lifetime heterogeneity, viscosity-lifetime calibration, analysis of SAXS/WAXS datasets, flickering spectroscopy, μ-DSC and analysis of MD simulations) please refer to the ESI.

**Statistical analysis.** Data is shown as mean ± S.D. Box plots display the 25–75% range, error bars represent ± S.D., median is shown by a horizontal line and mean by a dot. Origin® software was used to perform one-way ANOVA test. *$p < 0.05$; **$p < 0.01$; ***$p < 0.001$.

## Data availability

The data that support the findings of this study are available from the corresponding author upon reasonable request.

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

## Acknowledgements

M.P.P. acknowledges the Engineering and Physical Sciences Research Council (EPSRC) and the British Heart Foundation (BHF) for the Doctoral Training Studentship (EP/L015498/1, RE/13/4/30184) from the Institute of Chemical Biology (Imperial College London). M.K.K. is grateful to the EPSRC for a Career Acceleration Fellowship (EP/I003983/1). C.D.L. acknowledges the UK HPC Materials Chemistry Consortium, which is funded by EPSRC (EP/R029431), for providing access to Young (which is part of the UK Materials and Molecular Modelling Hub for computational resources, MMM Hub, which is partially funded by EPSRC (EP/T022213)), a Tier-II high performance computing resource, for the molecular dynamics simulations reported in this manuscript, R.L.R.D.C acknowledges the support by the Biotechnology and Biological Sciences Research Council (BB/T008709/1) via the London Interdisciplinary Doctoral Programme (LIDo).

## Author contributions

M.P.P. and A.V. performed the experiments, I.L.D. synthesized the molecular rotors and E.L. synthesized POPC-OOH. R.L.R.C and C.D.L. performed M.D. simulations. M.P.P., R.L.R.D.C and C.D.L. analyzed the data. All the authors contributed to designing the experiments, discussions and writing the manuscript. N.J.B. and M.K.K. supervised the project.

## Competing interests

The authors declare no competing interests.
