## [Peer Review File · Communications Chemistry]

Reviewers' comments:

Reviewer #1 (Remarks to the Author):

The work by Paez-Perez et al. investigates the effects of lipid peroxidation on the elastic and viscous properties using several complementary methods. While the overall topic is interesting and of broad interest, there are several points that should be clarified and discussed in more detail.

It is interesting that the authors see that the membrane rigidity decreases and the viscosity increases upon lipid peroxidation, which disorders the membrane; however, one of my major concerns is the authors' discussions of the 'elasticity and viscosity uncoupling' throughout the manuscript. While there are several theoretical works that have considered the viscoelastic properties of membranes, the current results separately study the elastic properties and the viscous properties, and these properties represent fundamentally different behaviors of the membrane. In classical elasticity theory, the elastic properties (i.e. the bending modulus and K_A discussed here) characterize the solid like properties of the membrane and its ability to completely recover its original shape after deformation (i.e. like a spring). The viscous properties represent the loss or dissipation of energy and are typically assigned to the fluid-like properties of the membrane (i.e. like a dashpot). These properties are not coupled. It is true that several reports have shown that both the elastic and viscous properties are correlated with the lipid structures, and that a more disordered membrane is often less rigid and less viscous, but this does not mean the elasticity and viscosity are directly coupled to one another as the elasticity does not determine the viscosity and vice versa. The current discussion of elastic and viscous properties is quite confusing and misleading, and the text should be revised.

Can the authors clarify what they mean by 'microviscosity'? The terms microviscosity and viscosity are used throughout the manuscript and it is not clear if these terms are used interchangeably or represent different membrane properties.

Another major result in the paper is the authors suggest ordered domains from in the oxidized membranes. The simulation results nicely show that the oxidized lipids preferentially associate, but I have a few questions that should be addressed regarding the experimental evidence for phase separation.

Have the authors considered the potential effects of H-bonding between their viscosity sensitive fluorophores and the peroxidized lipids? In their simulations, they discuss H-bonding between the -OOH groups in the oxidized tails and ester groups in the lipid heads. There are also ester groups in the BC10 and BC6++ dyes that could potentially form H-bonds with the peroxidized lipids. How would this affect the measured fluorescence lifetime? I would naively expect that specific interactions (such as H-bonds) could slow down the molecular rotation compared to the free dye? In which case, could the appearance of regions of different viscosities be due to specific interactions between the dyes and lipids and not phase separation? Similarly, the authors mention that they calculate the viscosity based on a previously published lifetime-viscosity calibration. Was this calibration performed for more than one type of solvent, including solvents that can form H-bonds? How does the changing polarity of the membrane with lipid peroxidation affect the lifetime?

The FLIM imaging results in Fig. 5 do suggest that the membranes containing peroxidized lipids are more heterogeneous, but I would have naively expected to see two distinct viscosity populations in the 50% POPC-OOH membrane? I might expect to see one population corresponding to 100% POPC and the other to 100% POPC-OOH if the proposed phase separation is due to clustering on the POPC-OOH lipids, especially if the dye does not preferentially segregate to one domain as the authors suggest? Can the authors comment on this?

Also concerning the experimental evidence for phase separation, the authors suggest that the increase in the width of the WAXS peak (Fig. 4c) was indicative of phase separation. While this broadening could suggest phase separation, it could also simply suggest that the membrane is more dynamic as seen in the SAXS results. Was the solvent background subtracted from the X-ray scattering data? As the membrane becomes more disordered and the peaks are less pronounced, it can be difficult to resolve peaks and the background signal from water. It would be good to check that the peak broadening is not a background effect.

The viscosity of three-dimensional bulk liquids often increases with H-bonding. Perhaps the increase in viscosity the authors see here is due to the increased H-bonding in the POPC-OOH lipids? H-bonding between the alkyl tail may explain why the viscosity increases despite the membranes becoming more disordered and less rigid with increasing lipid peroxidation. It may be interesting to plot the viscosity (Fig. 2b) as a function of GP value (Fig. 2d) to highlight the trend in viscosity with lipid order. Also, have the authors compared the trends present results for peroxidized lipids with other membranes known to form H-bonds in literature (e.g. sphingomyelins)?

Reviewer #2 (Remarks to the Author):

Perez-Perez et al. present experimental and simulation data for mixtures of POPC and an oxidized lipid, POPC-OOH. They find that the oxidized lipid increases membrane viscosity while decreasing membrane bending stiffness. This is unusual because viscosity and bending rigidity are typically strongly correlated, each increasing with increasing order of the lipid chains. I am impressed with the variety of biophysical techniques the authors have used to gain insight into the underlying mechanism for the decoupling of viscosity and bending stiffness. In particular, the MD simulations identify an interesting feature of POPC-OOH—namely the tendency of the oxidized sn2 chain to snorkel to the aqueous interface—that could certainly influence the viscosity. However, my major criticism lies with the authors' conclusion that mixtures of POPC/POPC-OOH, as well as single-component POPC-OOH bilayers, separate into coexisting liquid-disordered and liquid-ordered phases at temperatures below ~ 50-55 C. There are two major problems with this interpretation. First, phase separation in the pure POPC-OOH membrane would violate the Gibbs phase rule (at fixed pressure, a single-component bilayer can only have at most two coexisting phases, and only then at a fixed temperature). Second, to my knowledge the liquid-ordered phase is found only in phospholipid/sterol mixtures. I therefore cannot recommend that the paper be published in its present form.

Additional issues for the authors' consideration:

1. I'm not at all convinced that the fluorescence data in Fig. S2 show evidence of a phase transition. The authors base their argument on the expectation that these data should be perfectly linear if the bilayers are in a single phase over the entire temperature range, but I can't think of any physical reason why that has to be the case. Smoothly varying, monotonic behavior is also consistent with the absence of a first-order phase transition that would instead appear as an abrupt change in the signal, and at least to my eye, the data in Fig. S2 don't show abrupt changes. If the authors disagree, they should attempt to justify this by fitting the data to appropriate models and performing appropriate statistical analyses.

2. Related to the previous point, did the authors attempt to look at this hypothesized phase transition with DSC? Unfortunately, the DSC data shown in Fig. S6 do not go higher than 10 C.

3. I'm confused by the conclusion that the oxidized lipid increases membrane order. While some of the measurements (viscosity and Laurdan GP) appear to show increased order with increasing POPC-OOH, others show a disordering effect: (1) in the SAXS data (Fig. 4a), the first minimum of the form factor progressively moves to higher q with increasing POPC-OOH, indicating a thinner and more disordered bilayer; (2) increased bending fluctuations indicated by the broadening of the SAXS structure factor peaks are consistent with increased fluidization of the bilayer; (3) the WAXS peak shown in Fig. 4c broadens with increasing POPC-OOH, indicating a larger angular distribution of chain orientations (i.e., more disordered chains); (4) DSC data in Fig. S6 show that the gel to fluid transition of POPC shifts to lower temperatures with increasing POPC-OOH, indicating a disordering effect; (5) MD simulations presented in Fig. 6 show that POPC-OOH decreases the membrane thickness and dramatically increases the average area per lipid, consistent with decreased packing density and a more disordered bilayer. The authors need to do a better job explaining how these apparently contradictory data can all be consistent with their interpretation.

Reviewer #3 (Remarks to the Author):

This manuscript seeks to resolve an apparent contradiction in extant literature about the oxidation-induced changes in biophysical properties of bilayer lipid membrane. Using a combination of experimental techniques (fluorescence-based lifetime and spectroscopic measurements, spectroscopy, and XRD measurements) and atomistic molecular simulations, the authors quantify changes in the viscous (i.e., fluidity) and elastic (i.e., elastic moduli) properties of (i) mixtures of unmodified POPC and peroxidized POPC-OOH membranes and (ii) in situ photooxidized POPC measurements. Using both LUV and GUV configurations, they find that the presence of oxidized species led to an increase in membrane viscosity and led to phase separation.

The topic is important, work carefully executed, and the findings are insightful. I recommend publication after the following questions have been considered by the authors.

1. There has been a series of papers by Malmstadt et al. explicitly examining products of photo-induced lipid peroxidation and their effects on membranes, which might be relevant to the present study. In particular, I recommend considering the findings reported in the following:

Shalene Sankhagowit, Shao-Hua Wu, Roshni Biswas, Carson T. Riche, Michelle L. Povinelli, and Noah Malmstadt. *Biochimica Biophysica Acta - Biomembranes*. 1838(10):2615-2624. 2014.

"Viscoelastic deformation of lipid bilayer vesicles." Shao-Hua Wu, Shalene Sankhagowit, Roshni Biswas, Shuyang Wu, Michelle L. Povinelli, and Noah Malmstadt. *Soft Matter*. 11:7385-7391. 2015.

"Low levels of oxidation radically increase the passive permeability of lipid bilayers." Kristina Runas and Noah Malmstadt. *Soft Matter*. 11(3):499-505. 2015.

2. I am confused why not also measure diffusional properties such as by FRAP, and correlate these with the microviscosity measurements. A clarification will address the concern for the broader readership.

3. In this same vein, why not directly image the domain formation and phase separation of oxidized and native lipids using phase-sensitive dyes.

4. Consider presenting a schematic showing the types of attractive interactions in -OOH derivatized lipids with their neighbors.

5. It is not clear how does the formation of condensed liquid-ordered type domain be supported by oxidation, which results in increased membrane area, increased permeability, and decreased stretching modulus? How do the latter support generation of membrane mechanical stress? A more detailed systematic description of these mechanistic steps (and any evidence supporting their existence) is needed.

Reviewer #1 (Remarks to the Author):

The work by Paez-Perez et al. investigates the effects of lipid peroxidation on the elastic and viscous properties using several complementary methods. While the overall topic is interesting and of broad interest, there several points that should be clarified and discussed in more detail.

It is interesting that the authors see that the membrane rigidity decreases and the viscosity increases upon lipid peroxidation, which disorders the membrane; however, one of my major concerns is the authors' discussions of the 'elasticity and viscosity uncoupling' throughout the manuscript. While there are several theoretical works that have considered the viscoelastic properties of membranes, the current results separately study the elastic properties and the viscous properties, and these properties represent fundamentally different behaviors of the membrane. In classical elasticity theory, the elastic properties (i.e. the bending modulus and K_A discussed here) characterize the solid like properties of the membrane and its ability to completely recover its original shape after deformation (i.e. like a spring). The viscous properties represent the loss or dissipation of energy and are typically assigned to the fluid-like properties of the membrane (i.e. like a dashpot). These properties are not coupled. It is true that several reports have shown that both the elastic and viscous properties are correlated with the lipid structures, and that a more disordered membrane is often less rigid and less viscous, but this does not mean the elasticity and viscosity are directly coupled to one another as the elasticity does not determine the viscosity and vice versa. The current discussion of elastic and viscous properties is quite confusing and misleading, and the text should be revised.

We thank the reviewer for critically reading our work and for their constructive comments.

We agree with the reviewer's comment that, indeed, elasticity and viscosity are two independent mechanical descriptors. The mechanical behaviour of lipid membranes is ultimately dictated by both the intermolecular interactions between the constituent lipid molecules and the larger scale membrane structure (particularly bilayer thickness). However, in most phospholipid membranes, it is observed that stronger lipid-lipid attraction increases the energy cost for membrane deformation (i.e. higher bending and stretching moduli) *and* reduces the diffusion coefficient of the lipid molecules. We previously demonstrated that diffusion parameters are linked to the membrane's microviscosity [Dent et al. PCCP 2015]. This commonly leads to a direct correlation between the two descriptors in both model and cellular membranes, as highlighted by reviewer 2 and also in the literature, e.g. [Steinkühler et al. Commun. Biol. 2019]. We agree that the term 'coupling' could be misleading and have replaced the expression 'viscoelastic uncoupling' with 'a loss of viscoelastic correlation' throughout the manuscript and highlighted in the introduction that such observation is a casual correlation and that such relationship is not necessarily universal in lipid membranes. It now reads:

Experimentally, it has been suggested that both quantities, K_A and η , are correlated, as they both depend on the intermolecular interaction between the membrane's lipids;²³ yet this relationship is not necessarily universal, and exceptions could be seen in membranes with a more complex structure.²⁴⁻²⁶

Can the authors clarify what they mean by 'microviscosity'? The terms microviscosity and viscosity are used throughout the manuscript and it is not clear if these terms as used interchangeable or represent different membrane properties.

We thank the reviewer for highlighting this point. The term "viscosity" refers to a bulk property of the membrane, i.e. how fast the membrane will deform under a given load, while "microviscosity" refers to the free volume within the lipid molecules, which is what is sensed by molecular rotors. For clarity, we have replaced the term "viscosity" by "microviscosity" throughout the manuscript and added the following sentence clarifying this difference:

We note that, while viscosity reflects the time-dependent deformation of a substance for a given load, the term microviscosity refers to the molecular mobility of the probe's local environment,³⁹ which in turn defines the in-plane membrane diffusivity.

Another major result in the paper is the authors suggest ordered domains from in the oxidized membranes. The simulation results nicely show that the oxidized lipids preferentially associate, but I have a few questions that should be addressed regarding the experimental evidence for phase separation.

Have the authors considered the potential effects of H-bonding between their viscosity sensitive fluorophores and the peroxidized lipids? In their simulations, they discuss H-bonding between the -OOH groups in the oxidized tails and ester groups in the lipid heads. There are also ester groups in the BC10 and BC6++ dyes that could potentially form H-bonds with the peroxidized lipids. How would this affect the measured fluorescence lifetime? I would naively expect that specific interactions (such as H-bonds) could slow down the molecular rotation compared to the free dye? In which case, could the appearance of regions of different viscosities be due to specific interactions between the dyes and lipids and not phase separation?

We utilised two molecular rotors in this work, BC10 and BC6++ (Fig 1) both of which contain an ether (not an ester) group, which is not considered a strong H-bond acceptor. However, to exclude the possibility of an interaction with peroxidized lipids, we examined our MD simulation results, specifically looking at possible H-bond interactions (Fig. S17 in the updated manuscript) which show that the primary interaction between the BODIPY dye and the POPC-OOH lipids is through their hydrocarbon tails. There is no indication of any hydrogen bonding between the BODIPY and the oxidized lipids.

Fig. S17: Interaction map between BC10 and POPC-OOH. The highlighted row, corresponding to the oxygen in the peroxide, shows a lack of significant bonding between the dye and the oxidized lipid.

Similarly, the authors mention that they calculate the viscosity based on a previously published lifetime-viscosity calibration. Was this calibration performed for more than one type of solvent, including solvents that can form H-bonds? How does the changing polarity of the membrane with lipid peroxidation affect the lifetime?

The calibrations used here for two molecular rotor dyes, BC10 and BC6⁺⁺ (Fig 1), were performed in methanol/glycerol mixtures, at different methanol/glycerol ratios and at different temperatures, as described in the references [Hosny et al. Chem. Sci. 2016, López-Duarte et al. Chem. Commun. 2014], cited in the ESI. We also note that previous work from our group [Vyšniauskas et al. PCCP 2017], examined the effect of solvent polarity on the BC10 calibration and found no significant polarity effect for viscosities above ca 80 cP. Specifically, it can be seen from the lifetime data measured for BC10 in methanol/glycerol and toluene/castor oil mixtures that within the studied viscosity values (>150cP), the experimental datapoints collapse onto the same curve, suggesting that BODIPY sensors are, indeed, reporting on their surrounding microviscosity, with low sensitivity to changes in polarity. We added a sentence in the manuscript clarifying the lack of BODIPY sensitivity to polarity at our working viscosity ranges:

We note that molecular rotor BC10 is localized in the hydrophobic core of the membrane;⁵⁵ as the rotor shows a polarity and temperature-independent responses within the studied viscosity range,⁶⁵ it is expected that changes in BC10 lifetime are truly representative of changes in the bilayer's microviscosity.

The FLIM imaging results in Fig. 5 do suggest that the membranes containing peroxidized lipids are more heterogenous, but I would have naively expected to see two distinct viscosity populations in the 50% POPC-OOH membrane? I might expect to see one population corresponding to 100% POPC and the other to 100% POPC-OOH if the proposed phase separation is due to clustering on the POPC-OOH lipids, especially if the dye does not preferentially segregate to one domain as the authors suggest? Can the authors comment on this?

We have performed additional imaging experiments to answer the reviewer's point, Figure S12 in the updated manuscript. Indeed, in the case of 50% POPC-OOH membranes we observe two distinct viscosity domains (see figure below, note, the colour-scale has been adjusted to increase the contrast).

The shorter lifetime corresponds to t_2 of ca 1.85ns, while the longer t_2 have a value of ca 2ns, corresponding to viscosities of ~166 cP and 205 cP, respectively, as seen in the histogram distributions below. We ascribe the low-viscosity areas to POPC-rich regions (corresponding viscosity values measured in LUVs were ~160 cP). The areas of higher viscosity seen in GUVs, 205 cP, are slightly less viscous than seen in pure POPC-OOH bilayers (~240 cP). It is possible that these regions contain a certain proportion of non-oxidized POPC, weakening the interactions between the oxidized lipids, which is reflected in the lower viscosity. Partitioning in phase separated mixtures can never be perfect and this is known to occur to a significant degree in archetypical phase-separated membrane compositions [Gu et al. JACS 2020], hence it is very likely that in our system, POPC molecules could reside within POPC-OOH regions, lowering their microviscosity compared to pure POPC-OOH.

A reference has been added in the main text to figure S12:

The wider lifetime histograms as well as a non-gaussian histogram shape was seen for both 50% and 100% POPC-OOH (Fig. S12).

Fig. S12: POPC-OOH increases lateral heterogeneity in model GUVs. (a) Example of FLIM images. (b) Histograms representing the average of the pixel-wise lifetimes in each of the images displayed in (a). Scale bar: 20 μm .

Also concerning the experimental evidence for phase separation, the authors suggest that the increase in the width of the WAXS peak (Fig. 4c) was indicative of phase separation. While this broadening could suggest phase separation, it could also simply suggest that the membrane is more dynamic as seen in the SAXS results. Was the solvent background subtracted from the X-ray scattering data? As the membrane becomes more disordered and the peaks are less pronounced, it can be difficult to resolve peaks and the background signal from water. It would be good to check that the peak broadening is not a background effect.

We confirm that solvent background subtraction was performed as the initial step in our analysis; so, we are confident that the analysed peaks arise from membrane scattering and believe the peak broadening is not a background effect. We agree that this result is not a definite proof for the presence of POPC-OOH induced clusters but, rather, supporting evidence for the increase of membrane

heterogeneity, which is further suggested by the other data presented in this work. We have added the following sentence to highlight this:

However, the WAXS peaks became significantly broader, predominantly towards larger q -values (corresponding to lower lipid spacing), which could be compatible with a wider area per lipid (APL) distribution, as would be expected in a bilayer displaying lateral heterogeneity.

The viscosity of three-dimensional bulk liquids often increases with H-bonding. Perhaps the increase in viscosity the authors see here is due to the increased H-bonding in the POPC-OOH lipids? H-bonding between the alkyl tail may explain why the viscosity increases despite the membranes becoming more disordered and less rigid with increasing lipid peroxidation. It may be interesting to plot the viscosity (Fig. 2b) as a function of GP value (Fig. 2d) to highlight the trend in viscosity with lipid order. Also, have the authors compared the trends present results for peroxidized lipids with other membranes known to form H-bonds in literature (e.g. sphingomyelins, SM)?

We fully agree with the reviewer's suggested interpretation that H-bonding involving the -OOH in the alkyl tails (Fig. S16 in the original manuscript, Fig. S20 in the updated version) is the driving force for an increase in viscosity despite the decrease in membrane rigidity. We also note the order parameter of the unsaturated tail (S_{cd}) shows a marked increase when hydroperoxidation occurs, which we believe is an indication of the increased H-bonding interactions with the -OOH group driving an increase in the local lipid packing around that region.

Following the reviewer's suggestion, we have also plotted the relation between membrane viscosity and GP value for POPC as a function of temperature, and for a range of POPC-OOH/POPC ratios. The results, shown in the figure below (new Fig. S3a in the updated manuscript), suggest the correlation between Laurdan's GP and membrane viscosity is not significantly affected by the presence of hydroperoxidized lipids, compared to a control experiment where the membrane order was systematically changed by altering the temperature, for pure POPC liposomes. In addition, we would like to point out that, because the viscosity/GP relationship is not significantly affected upon the addition of POPC-OOH, such observation could be considered extra evidence for the lack of significant H-bond interaction between -OOH and BODIPY rotors, as otherwise we would have expected the relationship between these two observables to be altered.

Regarding the reviewer's suggestion of comparing lifetime trends in SM bilayers, we believe such comparison will not be fully representative, as previous studies have shown SM at room temperature exhibits a (rippled) gel phase [Shaw et al. *Soft Matter* 2012], in contrast to the fluid phase corresponding to the POPC/POPC-OOH system. Instead, we attempted to investigate the effect of electrostatic interactions between lipids and the rotor by measuring BC10 fluorescence and Laurdan GP in membranes containing charged PG and TAP headgroups. The results, shown in the figure below (new Fig. S3b in the updated manuscript), indicate a similar response regardless of the presence of charged lipids, thus, further supporting the hypothesis that membrane charge/electrostatic interactions have a negligible effect on BC10 compared to its sensitivity to the bilayer's microviscosity.

These considerations have been included in the main text as:

We note that the relation between Laurdan's GP and microviscosity does not appear to be significantly affected by the presence of lipid peroxides compared to standard temperature-induced changes in the membrane's structure (Fig. S3a), nor does the presence of charged lipids affect the readout from the BC10 probe (Fig. S3b).

Fig. S3: The relationship between the BC10 rotor readout and Laurdan's GP. (a) The correlation between GP and microviscosity is similar in pure POPC bilayers at different temperatures (black trace and dots) and in POPC liposomes containing increasing amounts of POPC-OOH (coloured data points). This suggests that the interaction between -OOH and BODIPY rotors have a negligible influence on the readout of the rotor. (b) The relationship between the membrane polarity (as measured by Laurdan's GP) and the BC10 fluorescence intensity (proportional to the membrane viscosity), measured at a range of temperatures for liposomes containing charged lipids, 1-palmitoyl-2-oleoyl-sn-glycero-3-phospho-(1'-rac-glycerol) (sodium salt) (POPG) and 1,2-dioleoyl-3-trimethylammonium-propane (chloride salt) (DOTAP). The similarity between the three responses suggests that the BC10 sensitivity is not significantly affected by changes in membrane charge.

Reviewer #2 (Remarks to the Author):

Perez-Perez et al. present experimental and simulation data for mixtures of POPC and an oxidized lipid, POPC-OOH. They find that the oxidized lipid increases membrane viscosity while decreasing membrane bending stiffness. This is unusual because viscosity and bending rigidity are typically strongly correlated, each increasing with increasing order of the lipid chains. I am impressed with the variety of biophysical techniques the authors have used to gain insight into the underlying mechanism for the decoupling of viscosity and bending stiffness. In particular, the MD simulations identify an interesting feature of POPC-OOH—namely the tendency of the oxidized sn2 chain to snorkel to the aqueous interface—that could certainly influence the viscosity. However, my major criticism lies with the authors' conclusion that mixtures of POPC/POPC-OOH, as well as single-component POPC-OOH bilayers, separate into coexisting liquid-disordered and liquid-ordered phases at temperatures below ~ 50-55 C. There are two major problems with this interpretation. First, phase separation in the pure POPC-OOH membrane would violate the Gibbs phase rule (at fixed pressure, a single-component bilayer can only have at most two coexisting phases, and only then at a fixed temperature). Second, to my knowledge the liquid-ordered phase is found only in phospholipid/sterol mixtures. I therefore cannot recommend that the paper be published in its present form.

We thank the reviewer for appreciating our work and their constructive comments. Regarding the reviewer's concerns, we would like to indicate that:

- i) Although our findings are apparently inconsistent with the Gibbs phase rule, one must consider that POPC-OOH can be found in two states (e.g. with the -OOH group either embedded within the hydrocarbon region or snorkelling towards the membrane's surface). While, thermodynamically, this should not allow formation of two phases, we believe that interconversion between the snorkelling and non-snorkelling configurations is slow, leading to kinetic trapping of the molecular states, which allows formation of the observed membrane heterogeneity. This is supported by our MD results that suggest approximately 40% of the POPC-OOH molecules display a snorkelling behaviour after the simulation time (Fig. S12 in the original manuscript, Fig. S15 in the updated version). In addition, apparent violations of the Gibbs rule are commonly seen in lipid vesicles, although poorly understood. For example, pure DPPC membranes display gel-fluid domain coexistence over a temperature range around the main transition, which has been attributed to finite-size effects and metastable states (e.g. Knorr et al. BBA-Biomembranes 2018).
- ii) We agree with the reviewer that the expression "liquid ordered phase" is usually related to sterol-containing lipid mixtures. Because the presence of POPC-OOH was able to increase the local order (as evidenced by the molecular rotor measurements) and decrease the long-range order (e.g. exemplified by the wider SAXS peaks indicative of increased membrane fluctuations) we described the high-viscosity regions as liquid ordered phase. However, we appreciate that this could be confusing so have followed the reviewer's advice and replaced the expression "liquid-ordered phase" with "more-ordered regions" throughout the manuscript.

Additional issues for the authors' consideration:

1. I'm not at all convinced that the fluorescence data in Fig. S2 show evidence of a phase transition. The authors base their argument on the expectation that these data should be perfectly linear if the bilayers are in a single phase over the entire temperature range, but I can't think of any physical reason why that has to be the case. Smoothly varying, monotonic behavior is also consistent with the absence

of a first-order phase transition that would instead appear as an abrupt change in the signal, and at least to my eye, the data in Fig. S2 don't show abrupt changes. If the authors disagree, they should attempt to justify this by fitting the data to appropriate models and performing appropriate statistical analyses.

We expect that, within a single phase, the molecular rotor readout is linear with respect to temperature, based on previous work in the group (e.g. Dent et al. PCCP 2015, Wu et al. PCCP 2013). Furthermore, some of our current ongoing studies hint at a clear relationship between microviscosity and the membrane's structural parameters, such as the area per lipid (measured by WAXS). Because it has also been reported in the literature that membrane's thermal area expansion coefficient changes between different phases of the same lipid, e.g. [T. Heimburg, BBA-Biomembranes 1998], we expect the rate of microviscosity change will differ between different membrane phases.

We appreciate that the change in slope is not as marked as expected from, for example, a gel-liquid transition but, arguably, the differences in structural properties of the membrane regions in the present study are also significantly smaller. To address the reviewer's comment, we performed a simple analysis to identify the presence of a change in the slope in the data of Fig. S2 (Fig. S5 in the updated manuscript). We evaluated the change in slope in two ways:

- i) We performed a linear fit to our traces over increasing temperature ranges (i.e. adding experimental values at higher temperatures). If, indeed, there was no change in the slope, the addition of these extra points should have had a minimal effect on the error metrics (e.g. R^2). We see no change in the fitting parameter for POPC membranes. On the contrary, the addition of extra points for POPC-OOH data decreased the fit quality above $\sim 50-55^\circ\text{C}$ (marked with a red arrow in the right column plots)
- ii) We also directly looked at the gradient (defined as $\frac{y(i)-y(i-1)}{x(i)-x(i-1)}$, $i \geq 2$). In the case where no change in slope is taking place, such value should remain approximately constant (as it is found for the pure POPC composition). On the contrary, in the case of POPC-OOH containing membranes a clear change in slope is seen at higher temperatures.

These two occurrences were observed in both T-scans presented in the updated Fig. S5, and this has been referenced in the main text as:

...intensity of POPC-OOH containing LUVs at increasing temperatures (Fig. S5). We expected these temperature-dependent curves to be linear if only one lipid phase is present. However, it could be seen that in the presence of POPC-OOH, a clear change in the slope...

Fig. S5: Investigating the effect of temperature on POPC-OOH containing LUVs. (a) Change in Laurdan's GP; the change in the slope with increasing temperature is evident for POPC-OOH-containing membranes. (b) Change in **BC10** fluorescence intensity (which is proportional to viscosity); for a liquid following Andrade's relationship ($\mu = Ae^{\frac{B}{T}}$), the trace is expected to be linear. However, in the case of POPC-OOH-containing membranes the change in the slope was observed. The points corresponding to a change in the slope are marked with arrows. It could be seen that both methods record deviations from linearity in the slopes at the same temperature: $\sim 50^\circ\text{C}$ for 50% oxidized lipid and $\sim 55^\circ\text{C}$ for 100% oxidized lipid. Further evidence for the change in slope is highlighting in (c,d) the plots showing the first derivative (defined as $\frac{y(i)-y(i-1)}{x(i)-x(i-1)}$, $i \geq 2$) and in (e,f) where the goodness-of-fit parameter R^2 is shown for a linear fit to all GP or fluorescence intensity data between 25°C and the temperature of the data point. If there was no change in slope the addition of additional points (at higher temperatures) should have a minimal effect on the error metric and thus the error would not significantly change – as we see when evaluating pure POPC membranes. On the contrary, if a change of slope is present, the addition of fitting points in these regions will decrease the fitting quality in regions with a different trend, and this will be reflected by a lower goodness of fit, as we indeed see above $\sim 50\text{--}55^\circ\text{C}$ for POPC-OOH containing membranes.

2. Related to the previous point, did the authors attempt to look at this hypothesized phase transition with DSC? Unfortunately, the DSC data shown in Fig. S6 do not go higher than 10°C .

We performed DSC scans on pure POPC-OOH samples from 20 to 70°C (heating rates of $0.25^\circ\text{C}/\text{min}$ and $0.5^\circ\text{C}/\text{min}$), but no transition was observed. This finding is consistent with a lipid arrangement where the lattice forces are too weak to drive a highly cooperative phase transition – in an analogous way to the disruptive effect of cholesterol incorporation in DPPC bilayers. [Mabrey et al. Biochemistry 1972] In the case of POPC-OOH bilayers, the weak lipid-lipid interaction (compared to bilayers in the gel phase) is reflected by the WAXS patterns observed, characteristic of membranes in the fluid phase.

3. I'm confused by the conclusion that the oxidized lipid increases membrane order. While some of the measurements (viscosity and Laurdan GP) appear to show increased order with increasing POPC-OOH, others show a disordering effect: (1) in the SAXS data (Fig. 4a), the first minimum of the form factor progressively moves to higher q with increasing POPC-OOH, indicating a thinner and more disordered bilayer; (2) increased bending fluctuations indicated by the broadening of the SAXS structure factor peaks are consistent with increased fluidization of the bilayer; (3) the WAXS peak shown in Fig. 4c broadens with increasing POPC-OOH, indicating a larger angular distribution of chain orientations (i.e., more disordered chains); (4) DSC data in Fig. S6 show that the gel to fluid transition of POPC shifts to lower temperatures with increasing POPC-OOH, indicating a disordering effect; (5) MD simulations presented in Fig. 6 show that POPC-OOH decreases the membrane thickness and dramatically increases the average area per lipid, consistent with decreased packing density and a more disordered bilayer. The authors need to do a better job explaining how these apparently contradictory data can all be consistent with their interpretation.

Indeed, the results appear counterintuitive, but this is actually one of the main findings of our work: although elasticity and viscosity have been found to be correlated in model and cell membranes [Steinkühler et al. Commun. Biol. 2019], the presence of lipid peroxides disrupts this relationship. To address the reviewer's comment, we have added the following schematic and text in the main manuscript text, in order to clarify how apparently conflicting results can be explained in light of the alkyl chains conformational change induced by the presence of -OOH groups.

Snorkelling of the -OOH containing alkyl chains towards the membrane surface will increase the area per lipid, but will also increase the molecular crowding within the bilayer and will promote H-bonding interaction between peroxidized lipids, leading to a decreased diffusivity and membrane fluidity. In addition, the emerging lipid-lipid interactions would stabilize the formation of lipid regions of higher microviscosity. Simultaneously, snorkelling of the peroxidized chains would decrease the chain-chain interactions at the bilayer's midplane, causing a decrease in membrane thickness and bending rigidity. Overall, this is reflected in the disruption of the canonical correlation between membrane elasticity and viscosity and promote the emergence of more-ordered lipid regions in otherwise single-component membranes.

Fig. 7 Effect of lipid peroxidation on membrane architecture. Snorkelling of the lipid tails leads to an increased lipid towards the interface packing, yet this causes the increase of the individual area per lipid. Concurrently, chain snorkelling creates a "void" at the membrane's midplane, which lead to a lower membrane thickness and elastic properties.

Reviewer #3 (Remarks to the Author):

This manuscript seeks to resolve an apparent contradiction in extant literature about the oxidation-induced changes in biophysical properties of bilayer lipid membrane. Using a combination of experimental techniques (fluorescence-based lifetime and spectroscopic measurements, spectroscopy, and XRD measurements) and atomistic molecular simulations, the authors quantify changes in the viscous (i.e., fluidity) and elastic (i.e., elastic moduli) properties of (i) mixtures of unmodified POPC and peroxidized POPC-OOH membranes and (ii) in situ photooxidized POPC membranes. Using both LUV and GUV configurations, they find that the presence of oxidized species led to an increase in membrane viscosity and led to phase separation.

The topic is important, work carefully executed, and the findings are insightful. I recommend publication after the following questions have been considered by the authors.

We thank the reviewer for their positive assessment of our work.

1. There has been a series of papers by Malmstadt et al. explicitly examining products of photo-induced lipid peroxidation and their effects on membranes, which might be relevant to the present study. In particular, I recommend considering the findings reported in the following:

- Shalene Sankhagowit, Shao-Hua Wu, Roshni Biswas, Carson T. Riche, Michelle L. Povinelli, and Noah Malmstadt. *Biochimica Biophysica Acta - Biomembranes*. 1838(10):2615-2624. 2014.
- "Viscoelastic deformation of lipid bilayer vesicles." Shao-Hua Wu, Shalene Sankhagowit, Roshni Biswas, Shuyang Wu, Michelle L. Povinelli, and Noah Malmstadt. *Soft Matter*. 11:7385-7391. 2015.
- "Low levels of oxidation radically increase the passive permeability of lipid bilayers." Kristina Runas and Noah Malmstadt. *Soft Matter*. 11(3):499-505. 2015

We thank the reviewer for highlighting this work that we did not previously cite. References to these studies have now been added to the main text (refs 34, 43 and 35, in the updated manuscript).

2. I am confused why not also measure diffusional properties such as by FRAP, and correlate these with the microviscosity measurements. A clarification will address the concern for the broader readership.

We agree that FRAP measurements have the potential to offer some useful insights, however, we were initially limited by availability of suitable equipment. In addition, these techniques measure fundamentally different parameters, molecular rotors sense the "free volume" (microviscosity) around them, while FRAP directly measures the diffusion of the fluorescent species. Additionally, the molecular rotor approach has two advantages which were important for this study: the imaging capability and the ability to follow viscosity changes dynamically, on the time scale of >seconds.

Comparisons between membrane viscosity and lipid diffusion have previously been reported for model membranes using BODIPY and FCS, resulting in similar outcomes [Wu et al. PCCP 2013; Dent et al. PCCP 2015] which has allowed us to make the diffusion coefficient estimates shown in the manuscript. We note that the diffusion coefficient estimated in the present work from the BC10 reported membrane microviscosity agrees with the changes found through MD simulations (Fig. 6h in the main text). Nevertheless, we have now carried out additional FRAP measurements for pure POPC and POPC-OOH GUVs doped with NBD-PE lipid, giving diffusion coefficients of 2.3 ± 0.4 and 1.3 ± 0.2 $\mu\text{m}^2/\text{s}$ respectively. These new experiments have now been added to the ESI (methods and Fig. S1) and referenced in the main text as:

This was further supported by the MD simulations described later (Fig. 6, Fig. S13) and by FRAP experiments on pure POPC and POPC-OOH membranes (Fig. S1).

Fig. S1: FRAP measurements of lipid diffusion in POPC and POPC-OOH GUVs. (a) FRAP traces showing the mean and s.d. of example recovery curves ($n=4$). (b) Estimated diffusion coefficient of NBD-PE lipids in POPC and POPC-OOH bilayers ($n>10$).

3. In this same vein, why not directly image the domain formation and phase separation of oxidized and native lipids using phase-sensitive dyes.

We assume that the reviewer is suggesting the use of a lipid probe such as Rhodamine-tagged PE, typically used for domain visualization in archetypical DOPC/DPPC/Cholesterol GUVs. We expect (and indeed measure) that the difference in properties between the more and less ordered POPC-OOH / POPC membrane regions is quite small and will not allow such probes to distinguish these domains selectively. In fact, this is supported by the equal partitioning of our BODIPY rotors within GUVs (the only difference between domains is revealed is by the probe's fluorescence lifetime and not its partitioning, *i.e.* brightness).

4. Consider presenting a schematic showing the types of attractive interactions in -OOH derivatized lipids with their neighbors.

We have included such schematic in Fig.7 (see below), emphasizing the interaction between -OOH and the lipid headgroup. In addition, such interactions are highlighted in Fig. S20 of the updated ESI.

5. It is not clear how does the formation of condensed liquid-ordered type domain be supported by oxidation, which results in increased membrane area, increased permeability, and decreased stretching modulus? How do the latter support generation of membrane mechanical stress? A more detailed systematic description of these mechanistic steps (and any evidence supporting their existence) is needed.

We thank the reviewer for highlighting the need for a more in-depth discussion, which has now been added to the main text (see below). In addition, we have incorporated an additional figure to clarify the proposed mechanism by which membrane elasticity is uncoupled from the bilayer's microviscosity.

Snorkelling of the -OOH containing alkyl chains towards the membrane surface will increase the area per lipid, but will also increase the molecular crowding within the bilayer and will promote H-bonding interaction between peroxidized lipids, leading to a decreased diffusivity and membrane fluidity. In addition, the emerging lipid-lipid interactions would stabilize the formation of lipid regions of higher microviscosity. Simultaneously, snorkelling of the

peroxidized chains would decrease the chain-chain interactions at the bilayer's midplane, causing a decrease in membrane thickness and bending rigidity. Overall, this is reflected in the disruption of the canonical correlation between membrane elasticity and viscosity and promote the emergence of more-ordered lipid regions in otherwise single-component membranes.

Fig. 7 Effect of lipid peroxidation on membrane architecture. Snorkelling of the lipid tails leads to an increased lipid towards the interface packing, yet this causes the increase of the individual area per lipid. Concurrently, chain snorkelling creates a "void" at the membrane's midplane, which lead to a lower membrane thickness and elastic properties.

REVIEWERS' COMMENTS:

Reviewer #1 (Remarks to the Author):

I thank the authors for their detailed responses. Their revisions have addressed my comments and questions.

Reviewer #2 (Remarks to the Author):

The authors put a lot of work into their response letter and for the most part, I'm satisfied with their answers and revisions. I would however recommend that they include their response to my question about the Gibbs phase rule in the paper.

Reviewer #3 (Remarks to the Author):

I believe the questions raised during the last round of review – especially those raised by reviewer 2 – merit a more elaborate discussion in the main manuscript. Two points are particularly important: (1) formation of phases in single-component membranes and (2) the apparent contradiction in the relation between bending rigidity and fluidity.

I recommend publication after the authors provide a complete context of these points, the apparent controversy, and their proposed explanation with relevant references within the main body of the manuscript.

Reviewer #1 (Remarks to the Author):

I thank the authors for their detailed responses. Their revisions have addressed my comments and questions.

We thank the reviewer for the positive assessment of our revised version.

Reviewer #2 (Remarks to the Author):

The authors put a lot of work into their response letter and for the most part, I'm satisfied with their answers and revisions. I would however recommend that they include their response to my question about the Gibbs phase rule in the paper.

We are pleased our answers were able to address the reviewer's comments.

We have now included a new paragraph addressing the apparent contradiction to the Gibbs phase rule in the main manuscript (highlighted in green in the revised version), which now reads:

We note that, although the presence of lipid clusters in single-component membranes is apparently inconsistent with the Gibbs phase rule, it must be considered that POPC-OOH is found in two states (e.g. with the -OOH group either embedded within the hydrocarbon region or snorkelling towards the membrane's surface). While, thermodynamically, this should not allow formation of two phases, we believe that interconversion between the snorkelling and non-snorkelling configurations is slow, leading to kinetic trapping of the molecular states, which allows formation of the observed membrane heterogeneity. This is supported by our MD results that suggest approximately 40% of the POPC-OOH molecules display a snorkelling behavior during the simulation time (Fig. S15). In addition, apparent violations of the Gibbs rule are commonly seen in lipid vesicles, although poorly understood. For example, pure DPPC membranes display gel-fluid domain coexistence over a temperature range around the main transition, which has been attributed to finite-size effects and metastable states.⁷⁹

Reviewer #3 (Remarks to the Author):

I believe the questions raised during the last round of review – especially those raised by reviewer 2 – merit a more elaborate discussion in the main manuscript. Two points are particularly important: (1) formation of phases in single-component membranes and (2) the apparent contradiction in the relation between bending rigidity and fluidity.

I recommend publication after the authors provide a complete context of these points, the apparent controversy, and their proposed explanation with relevant references within the main body of the manuscript.

We thank the reviewer for the positive comments.

The importance of single-phase phase separation is now stressed in a new paragraph (highlighted in green in the revised version), which now reads:

We note that, although the presence of lipid clusters in single-component membranes is apparently inconsistent with the Gibbs phase rule, it must be considered that POPC-OOH is found in two states (e.g. with the -OOH group either embedded within the hydrocarbon region or snorkelling towards the membrane's surface). While, thermodynamically, this should not allow formation of two phases, we believe that interconversion between the snorkelling and non-snorkelling configurations is slow, leading to kinetic trapping of the molecular states, which allows formation of the observed membrane heterogeneity. This is supported by our MD results that suggest approximately 40% of the POPC-OOH molecules display a snorkelling behavior during the simulation time (Fig. S15). In addition, apparent violations of the Gibbs rule are commonly seen in lipid vesicles, although poorly understood. For example, pure DPPC membranes display gel-fluid domain coexistence over a temperature range around the main transition, which has been attributed to finite-size effects and metastable states.⁷⁹

Regarding the apparent contradiction between bending rigidity and fluidity, included the following sentence to highlight the importance of such finding.

... otherwise single-component membranes (Fig. 7 The change in the relationship between membrane viscosity and bending rigidity suggests a shift in the balance between the inter-leaflet and intra-leaflet lipid interactions; hence, we anticipate the drastic effect of lipid peroxidation ...